



# Daily water-level variations of supraglacial lakes in the southern Inylchek Glacier, Central Asia

Naoki Sakurai[1], Chiyuki Narama[2], Mirlan Daiyrov[3], Muhammed Esenamanov [3], Zarylbek Usekov [3], Hiroshi Inoue[4]

[1]Niigata University, Graduate School of Science and Technology, Niigata, Japan
[2]Niigata University, Program of Field Environmental Research, Niigata, Japan
[3]Central Asian Institute for Applied Geosciences (CAIAG), Bishkek, Kyrgyzstan
[4]National Research Institute for Earth Science and Disaster Resilience (NIED), Tsukuba, Japan

*Correspondence to*: Naoki Sakurai (n.sakurai1001@gmail.com)

**Abstract.** To better understand the storage in and drainage through supraglacial lakes and englacial conduits, we investigated the daily water-level variations of supraglacial lakes on the southern Inylchek Glacier in Kyrgyzstan. To examine these variations, we used daily aerial digital images over three years (22 July-15 August 2017, 8-29 July 2018, and 12-19 July 2019) from an unmanned aerial vehicle (UAV) that were converted to digital surface models (DSMs) and ortho-
images. Our main results are as follows. 1) When one lake drained, the water levels of other lakes might simultaneously increase, indicating that drainage water is shared with several lakes through a main englacial conduit. In one drainage event, a branch englacial conduit clearly connected to a main englacial conduit. 2) Sometimes, several lakes discharged simultaneously, indicating that several lakes had connected to a main englacial conduit that had opened. Such a case can cause larger-scale drainage than that from the opening of a branch englacial conduit. 3) Several lakes discharged twice in the
same year, each time through a different conduit, indicating that the main englacial conduit can be abandoned and reused. 4) In some lakes, the water level gradually increased with nearly the same increase rate just before drainage. Such an increase may be an indicator of imminent lake drainage.

## 1 Introduction

In Asian mountain regions, large-scale drainage and flooding occurs from supraglacial lakes on debris-covered glaciers. For
example, during May–June 2008 the Hunza Valley of the Karakoram had four large drainages from the terminus of the Ghulkin Glacier. These drainages destroyed homes and four irrigation channels in Borit Village as well as temporarily closed the Karakoram Highway (Richardson et al., 2009). On 29 April 2009 in the Lunana region of northwestern Bhutan, a sudden large discharge occurred through englacial and subglacial conduits at the Tshojo Glacier. The people of Punakha Town, located 70 km downstream of the glacier, were displaced by this flood (Komori et al., 2012). On 12 July 2016, the Lhotse
Glacier in the Khumbu region of eastern Nepal had large-scale drainage from a large supraglacial lake (Rounce et al., 2017). Later, from April to July 2017 on the Changri Shar Glacier, a branch of the Khumbu Glacier, a supraglacial lake of 180,000





m² rapidly formed, completely discharging soon thereafter in mid-July (Miles et al., 2018). In none of these cases did researchers find evidence for a large proglacial lake prior to the drainage, but in all cases large supraglacial lakes had drained. For example, at the Tshojo Glacier drainage event, the appearing and vanishing of a large supraglacial lake was confirmed

after drainage (Yamanokuchi et al., 2011). In addition, a large outlet of englacial conduit through which drainage water flowed was observed at the glacier terminus. The drainage was considered to have been caused by stored water in the supraglacial lake as well as in the englacial and subglacial conduits (Komori et al., 2012). Thus, recent large-scale drainages are related to supraglacial lakes and englacial conduits on debris-covered glaciers.

Supraglacial lakes on debris-covered glaciers form by water flowing into a basin or depression through englacial or surface

channels, and then continue to develop until a connecting englacial conduit opens (Benn et al., 2017; Narama et al., 2017). Development of a supraglacial lake depends on the surface gradient and flow velocity (Reynolds, 2000; Liu et al., 2015; Miles et al., 2016) as well as the conditions of the glacier mass balance (Benn et al., 2012). Concerning the effects of supraglacial lakes on the underlying glacier, they accelerate glacier ablation (Sakai, 2001; Benn et al., 2012), particularly those with high turbidity water (Takeuchi et al., 2012).

The appearing and vanishing of supraglacial lakes can occur when englacial conduits close and reopen, and such activity occurs due to glacier flow, freezing of stored water, and from a deposition of ice and debris due to tunnel collapse (Gully and Benn, 2009; Narama et al., 2018).

Seasonal variations of supraglacial lakes have been reported in recent years (Watson et al., 2016; Miles et al., 2016; Benn et al., 2017; Narama et al., 2017). For example, in the Langtang Valley of central Nepal, supraglacial lakes expand in the pre-

monsoon period due to snow melt, and then shrink due to the opening of englacial conduits that occur from increasing precipitation and snow-ice melting in the monsoon period (Miles et al., 2016). In the Khumbu region of the central Himalaya, the surface area of supraglacial lakes is larger in summer due to melt and precipitation, and expansion of englacial channels is caused by rising water temperature (Watson et al., 2016).

In the study area of the Tien Shan, satellite data in 1999–2015 show the number of supraglacial lakes to vary seasonally.

This number begins to increase in April, reaches a maximum from May to June, and then decreases from June to July (Narama et al., 2017). The increase in number from April to June is caused by melt, and the later decrease in number is caused by connections of the supraglacial lakes to the englacial conduit network. The timing of maximum number of lakes varies year by year, with the variation being larger on glaciers with a larger debris-covered area (Narama et al., 2017).

Daily variations of water levels have been measured in supraglacial lakes using water-level loggers with atmospheric and

water pressures (Narama et al., 2017; Miles et al., 2017), but the relationship between supraglacial lakes and englacial conduits remains unclear. In this study, to better understand the storage in and drainage through supraglacial lakes and englacial conduits, we investigated the daily water-level variations of supraglacial lakes on the southern Inylchek Glacier in Kyrgyzstan in 2017, 2018, and 2019 using an unmanned aerial vehicle (UAV).



## 2 Study area

The study area is the southern Inylchek Glacier in the central Tien Shan mountains. This glacier is the largest in Kyrgyzstan (about 500 km², 60.5 km long), the debris-coverd zone stretches over 22 km upward from the glacier terminus (Shangguan et al., 2015; Narama et al., 2017), and medial moraines from several branch glaciers. The average annual mass balance is −0.28 ± 0.46 m w.e. a⁻¹ during 1999-2007 (Shangguan et al., 2015), with surface-flow velocities in its upper part being faster than those on its lower part (Nobakht et al., 2014; Shangguan et al., 2015). The estimated maximum thickness is 380 m, and the

mean thickness of its debris-covered part is 136 m (Pieczonka et al., 2018). Many supraglacial lakes have developed on its debris-covered area with an annual seasonal drainage cycle (Narama et al., 2017).

The closest meteorological stations are the Tien Shan Station (TS) (78.2°E, 41.9°N, 3614 m asl., data from 1960–1997) and the Koiliu Station (K) (79.0°E, 42.2°N, 2800 m asl., data from 1960–1990; Fig. 1). The annual precipitation at TS and K stations are 279 mm a⁻¹ and 311 mm a⁻¹, respectively (Reyers et al., 2013). About 75% of the annual precipitation occurs

from March to September with peak precipitation occurring in June–July. According to observation data at TS, the annual average air temperature is −7.7 °C, with the lowest average temperature being −21.8 °C in January, the highest being 4.3 °C in July (Osmonov et al., 2013; Shangguan et al., 2015).

## 3 Methods

### 3.1 Field survey

Our field survey ran during the summer in 2017, 2018, and 2019 in the middle part of the southern Inylchek Glacier near the Lake Merzbacher base camp (Fig. 1). Aerial photographs were taken using a UAV (Phantom-3 and -4: DJI) each day from 22 July 2017 to 15 August 2017, from 8 July 2018 to 29 July 2018, and from 12 July 2019 to 19 July 2019 to determine water-level variations of supraglacial lakes. The area of the UAV survey was 2 × 2 km² in 2017 and 2 × 2.5 km² in 2018–

2019. The photo overlap and sidelap were 80% and 65% respectively. The survey time was 05:30–07:00 am, as this time period was hardly affected by weather and sunlight. To make ortho-images and DSMs, we used GCPs (ground control points) in 2017 (Fig. 2a), in 2018 (Fig. 2b), and in 2019 (Fig. 2c) that had been measured by a GNSS survey for 30 minutes with Trimble GeoExplorer6000. We obtained an accuracy with >20 cm at GCPs positions through differential post-processing of our GPS data using data from Kyrgyz GNSS reference station. We also set water-level loggers (Hobo U20)

with an interval of 1 hour and time-lapse cameras (Brinto) with an interval of 1 day from 2016 to 2019 (Fig. 2).

### 3.2 GIS analysis

We created ortho-images and digital surface models (DSMs) with 15-cm resolution from UAV aerial images. To make the ortho-images and DSMs, we used Pix4D Promapper of SfM (structure form motion) software (Fig. 2; Immrezeel et al., 2014;





Piermattei et al., 2015). Workflow of SfM software was almost automatic without setting of GCPs which we measured (Fig.

2). The daily water-levels were derived from these UAV DSMs. The acquisition of water levels is the following.

1) We set a ortho-image and DSM data as the standard DSM each year (for example, we set a DSM on 18 July as standard data in 2018).

2) Since the glacier surface flows at a speed of about 4m/month, all other ortho-images and DSM data were fitted to the standard data by shifting them with the use of the 'georeference' function in ArcGIS.

3) We made lake polygons manually for each day using each shifted ortho-images.

4) We converted the lake polygon to points at 15-cm intervals.

5) Water-level elevations of point data were extracted from the standard DSM. We used the average elevations of points (more than 500 points) excluding 45% of the highest and lowest values. The change of water volume also was calculated based on the standard DSM with the use of the 'Cut-Fill' function in ArcGIS.

Lake area variations outside the survey period were determined from Landsat8/OLI, Sentinel-2, and PlanetScope satellite images. For these satellite data, we manually delineated lake outlines, as in point 3 above. All satellites covered the survey period in 2017−2019.

**3.3 Accuracy of water-level variation**

Our water-level data was tested against direct measurements from water-level loggers. The water-level logger measurements came from two lakes in 2017 (Nos. 8, 10; Fig. 2a) and three lakes in 2018 (Numbers=Nos. 4, 7, 8; Fig. 2b). The water-level logger of lake (No. 4) in 2017 and lake (No. 10) in 2018 could not acquire water-level variations, because water-level logger was not covered by lake water (No. 4), and water-level logger of lake (No. 10) could not get the large variations within UAV survey periods. The direct measurements minus the UAV-DSM values were −26.8 ± 11.9 cm from 22 July to 15 August (No.

8) and −2.9 ± 7.1 cm from 23 July to 15 August (No. 10) in 2017, and 7.8 ± 5.6 cm from 8 July to 17 July (No. 4), −9.0 ± 10.1 cm from 8 July to 17 July (No. 7), and −7.3 ± 7.2 cm from 8 July to 17 July (No. 8) in 2018. The average difference of five lakes was −7.6 ± 8.4 cm. These accuracies are enough to analyze lake-level variations, because lake levels change from several meters to 20 m maximum.

**4 Results**

**4.1 Number and distribution of supraglacial lakes in mid-July 2017-2019**

Figure 2a–c shows UAV ortho-images in mid-July 2017, 2018, and 2019 (location in Fig. 1). Locations of the supraglacial lakes, shown in blue, are similar in all three years (Fig. 2a, b, c). All lakes were turbid. The number of lakes (>100 m²), all within the same overlapped area of the ortho-images 2017, 2018, and 2019, are 37 on 23 July 2017, 42 on 15 July 2018, and 48 on 16 July 2019, with the distribution falling sharply with increasing area (Fig. 2d). The number of lakes over 1000 m² in



area is 9 in 2017, 17 in 2018, and 16 in 2019. The largest of these supraglacial lakes (>5000 m²) were located in the central part of the glacier. The number and area of lakes in 2018 and 2019 are larger than those in 2017.

## 4.2 Storage and drainage phenomena

### 4.2.1 Drainage event in July 2017

Consider the storage and drainage behavior of the nine lakes (Nos. 1, 8, 9, 10, 11, 12, 13, 14, 16) marked in Fig. 3a. Their
water-level elevations in summer 2017 range from 3307 to 3321 m, but each one varies during 22 July to 15 August 2017 (Fig. 3b). For example, the water level of the largest lake (No. 8) increases gradually by 0.3 m (2723 m³) until 25 July 2017, begins to drop sharply by 9.8 m (62293 m³) from 26 July to 4 August, and the water level did not change after that. The water volume of lake was calculated based on difference of water levels for two days using Cut-Fill method in ArcGIS. The water level of the small lake (No. 9) drops 2 m (328 m³) from 24 July to 30 July, then the water level did not change. Before-
and-after ortho images in Fig. 4 (25 and 26 July 2017) show the change in surface conditions from drainage of lakes 8 and 9. The two lake areas clearly shrink after drainage, and a conduit with ice exposed parts was observed on the glacier surface between them during the field survey. That exposure parts were caused by erosion from the drainage water flowing from lake 8 (Fig. 4b).

Lake 1 discharges twice during the 25-day observation period, with its water level declining by about 10 m after each
drainage (Fig. 3b). Specifically, the water level declines 9.9 m (1378 m³) on the first drainage event on 24 July 2017, then the water-level increases by 7.5 m (738 m³) on 25 July (Fig. 3b). On the second drainage event, the water level drops 10.7 m (1756 m³) on 5 August. Then, four days later, the lake recharges, increasing by 8.3 m (942 m³) over one day. All three lakes (1, 8, 9) show a gradual increase of water level before drainage.

 The water level of lake 16 also increases gradually, but it does not drain during the survey period. The remaining lakes (Nos.
10, 11, 12, 13, 14) maintain a stable water-level.

### 4.2.2 Drainage event in May 2018

Figure 5a shows water-level variations based on water-level data loggers from 1 May to 18 July 2018 for lakes 8 and 10, as well as from 8 to 18 July for lakes 4 and 7. Lake 10 discharged at 1:00 AM on 7 May, followed by lake 8 at 5:00 AM on 13
May (GTM +9:00). (Later, lakes 4 and 7 discharged at the same time on 17 July.) Figure 5b shows time-lapse photos taken at lake 4 at 14:00 (GTM + 09:00) on 12 May and 14:00 on 13 May showing that this lake drained at about the same time as that of lake 8. Additional images in Fig. 5c,d show the drainage of lakes (Nos. 1, 4, 5, 7, 8) during 10–13 May. The yellow rectangles of Fig. 5c show newly exposed ice on 13 May, indicating the two drainage routes.





### 4.2.3 Drainage event in July 2018

For July 2018, we examined the 14 lakes marked in Fig. 6. The seven lakes labeled with orange highlighting in Fig. 6a,b (Nos. 1, 4, 5, 7, 8, 14, 15) simultaneously discharged on 17–18 July (Fig. 6c). The water-level data loggers at lakes 4, 7 and 8 recorded these drainages at around 12:00–15:00 on 17 July (Fig. 5a). In contrast, the lakes highlighted in white (Nos. 2, 10, 11, 12, 13) do not change during this period in July, whereas those highlighted in green (Nos. 3, 6) change, but not during the simultaneous drainage event (Fig. 6d).

The cases of lakes 3, 4, 5and 7 are examined more closely in Fig. 7. For lake 7, the level drops about 14 m during the 17 July drainage event (Fig. 7a,b). For lake 4, field inspection on 18 July showed a large englacial conduit after drainage, which dropped about 20 m during the same event (Fig. 7c). For Lake 5, the large englacial conduit was exposed after drainage on 18 July (Fig. 7d). The englacial conduit had developed vertically. Lake 3 drained a day earlier, showing a drop of about 5 m (Fig. 7e).

Before the drainage event, the water levels of lakes 1, 4, 7, and 8 gradually increased by 0.6–2 m. In contrast, the level of lake 5 decreased gradually by 2.5 m (7999 $m^3$) from 8 to 12 July and then maintained the same level before drainage on 17 July. Then, after the drainage event, lakes 4, 5, and 7 recharged, with two of them (Nos. 4, 7) discharging again (23, 25 July). According to PlanetScope satellite images, lake 5 also discharged again, but on a later date (7–9 August).

Of the seven lakes that drained on 17 July, five (Nos. 1, 4, 5, 7, 8) also drained on 13 May. The two lakes that did not discharge in 13 May (lakes 14, 15) also recharged to within 0.2 m of the water level before the drainage. These two lakes have the lowest elevations of all seven lakes in the event. This result shows drainage route was changed to the direction of lakes 14, 15 after 13 May drainage event.

For drainage on 17 July, the water flowed through an englacial conduit with some parts exposed on the surface. For instance, in the case of lake 15, Fig. 8 shows ice walls that were exposed on the surface due to drainage water on 17 July (location in Fig. 2b). The drainage route with exposed ice wall in July was different to the position of drainage route with exposed ice wall in May. Those difference indicates which drainage routes changed between May and July.

Of the lakes that did not respond to the drainage event on 17 July, the five with white highlight in Fig. 6a,b (Nos. 2, 10, 11, 12, 13) had a constant level through July (Fig. 6d). Lake 3 instead discharged on 16 July, and lake 6 discharged on both 10 and 15 of July (Fig. 6d). The water levels of lake 3 increased 0.6 m from 8 July to 16 July before draining, whereas lake 6 increased 0.5 m from 8 July to 9 July before draining the first time, and then gradually 4.2 m from 10 July to 14 July before its second drainage.

The large volume of water drained from lake 3 (77700 $m^3$; see Fig. 7d) left visible erosion marks. For example, the blue and red boxes in Fig. 9a, b show exposed ice walls of an englacial conduit with some parts exposed on the surface due to erosion of drainage water from the lake. The exposed ice was also observed in a field survey (Fig. 7e). The blue box shows lake water flowed through a surface channel to lake 4.





### 4.2.4 Drainage event in May and July 2019

According to Landsat8 (21 May 2019) and Sentinal-2 (28 May 2019), and PlanetScope (9, 11, 12 July 2019) images,
drainage in 2019 occurred at lakes 1, 4, 5, 7, and 8 during 21–28 May, and lakes 1, 4, 7, and 8 during 9–12 July. These drainage events resembled those in 2018; however, lakes 14 and 15 did not discharge in May and July.

After the first drainage in May 2019, exposed ice was observed on the same route as that in May 2018. This finding indicates that the first drainage event followed the same route in both years. However, the second drainage route in July 2019 is unclear. Also, this second drainage occurred earlier than that in 17 July 2018. After the drainages in July 2019, each lake
discharged at later, but different, times. Lakes 7 and 4 discharged again on 14–15 July and 24–27 July, respectively, based on UAV and satellite data.

## 5 Discussion

### 5.1 Connecting to main or branch englacial conduits

Seven lakes (Nos. 1, 4, 5, 7, 8, 14, 15) drained at the same time on 17 July 2018, indicating that they shared one or more englacial conduits despite being (Fig. 2c). Five of the lakes (Nos. 4, 5, 7, 14, 15) recharged after the drainage event, and then three of them (Nos. 4, 5, 7) discharged again later, though at different times (Fig. 6c). The different times of discharge indicates that the three lakes (Nos. 4, 5, 7) no longer shared an englacial conduit after the earlier drainage event, indicating that some part of the englacial conduit system changed.

The storage and drainage of supraglacial lakes is caused by closure and opening of englacial conduits (Benn et al., 2012). The closure of an englacial conduit can be caused by freezing of storage water or a deposition of ice and debris due to tunnel collapse (Gully and Benn, 2009; Narama et al., 2018). The recharging of the three lakes (Nos. 4, 5, 7) occurred by a closing of the englacial conduit. However, the conduit was not closed completely because the lakes discharged again within a few days. Such temporary blockages may be due to external or internal collapses, or to transported debris or ice. Fountain and
Walder (1998) reported that englacial conduits develop vertically from crevasses, and then form complex branching networks by uniting with other englacial conduits. These englacial conduits are complex shapes such as curving and branching channels (Gulley and Benn, 2007; Gulley et al., 2009). Thus, such a network consists of a main conduit to which branch conduits connect from various lakes. The discharge event of each lake after recharging may be due to an opening of a branch englacial conduit connecting to the main englacial conduit.

The above behavior is consistent with the englacial network sketched in Fig. 10. The first drainage of lake 5 on 8 July 2018 was caused by the circled branch englacial conduit in (a). This drainage was not complete, and the lake's water level was maintained until the multiple drainage event on 17 July. Then, the latter drainage event drained 1, 4, 5, 7, 8, 14, and 15 when the main englacial conduit opened as shown in (b) and (c). This opening may have been caused by an increase in water





pressure, at least partly due to the release of water from lake 5, water supplied to lake 4 from lake 3 on 16 July, and an
increase of meltwater on the glacier surface. Then the recharging of lakes 4, 5, and 7 are likely due to temporal closure of
each branch englacial conduit by debris and ice deposits (d). Subsequent discharging of these lakes then occurred due to
branch conduits opening at different times as in (d) and (e). In this way, the branch englacial conduits cause small-scale
drainage, whereas opening of the main englacial conduit can cause large-scale drainage.

A sudden recharge also suggests the presence of subsurface channels from other lakes. For example, in July 2017, after each
of two drainages, the level of lake 1 increased 7.5 m and 8.3 m over one day. The sudden and large increase indicates that
inflow occurred from one or more lakes through an englacial network.

## 5.2 Abandonment and reuse of englacial conduits

Of the seven lakes that drained simultaneously (Nos. 1, 4, 5, 7, 8, 14, 15) on 17 July 2018, the first five of them also drained
simultaneously two months earlier (13 May). Thus, the drainage route of the first five lakes did not pass through the last two
lakes (Nos. 14, 15) in May, yet did so in July (Figs. 5c, 8). This indicates a change in the englacial conduit between May and
July. Specifically, the five lakes (Nos. 1, 4, 5, 7, 8) and the two lakes (Nos. 14, 15) did not connect through the main
englacial conduit (Fig. 10f). But in July, the last two lakes connected to the first five through a formerly englacial channel
now visible at the glacier surface.
In May and July 2019, four of the same five lakes (Nos. 1, 4, 7, 8, but not No. 5) drained simultaneously, similar to their
drainage events in May and July 2018. This repeating behavior indicates that the englacial conduit in 2018 and 2019 had the
same network connections. As lake 5 did not drain in July 2019, its branch englacial conduit probably did not connect to
main englacial conduit.

The observed behavior is consistent with the distribution of englacial conduits shown in Fig. 11. The branch englacial
conduits in orange connect lakes or basins to a main englacial conduit in red. Part of the main englacial conduit changed
between May and July 2018, as shown with the dashed red curve, and the main englacial conduit changed back again the
next year in May 2019. These changes of main englacial conduit are consistent with the repeat cycles of abandonment and
reuse reported by Benn et al. (2012; 2017). Consistent with such a change, we observed an englacial conduit that split into
two channels (Fig. 7f; photo locations in Fig. 2a). Thus, englacial conduits generally could contain several routes and
continually change channels.

## 5.3 Phenomena of simultaneous drainage events

The DSM and water-level loggers showed the water level of several lakes to increase gradually before drainage, consistent
with previous studies (Miles et al., 2017; Narama et al., 2017). For example, Fig. 12a shows the increase in water level of
lakes 1, 4, 7, and 8, just before discharge on 17 July 2018. Three of these lakes increase at nearly the same relatively low rate





despite their water levels being at different elevations before drainage (Fig. 6c). This behavior indicates that the opening of an englacial conduit may have been caused by an increase in water pressure in the conduit due to an increase in the amount of water. However, if those lakes were connected to same englacial conduit, water-levels of those lakes become the same level, like a communicating vessels.

To further examine the cause of the opening, we examine the relationships between differences of the water level and the horizontal distance from lake 8, the lowest elevation lake in the group. Referring to Fig. 12c, the orange points (Nos. 1, 4, 5b, 7, 8) are lakes that drained simultaneously. The blue points are lakes unrelated to the same drainage event (Nos. 2, 3, 10, 11, 16, 17, 18, 19, 20, 21, 22). The black point (No. 5a) indicates the water level of lake 5 before the first drainage on 8 July 2018, whereas 5b marks the water level before the second drainage on 17 July 2018.

In Fig. 12c, we also added a hydraulic gradient line (orange, dashed) because recent glacier studies have found it useful to examine the hydraulic potential (Benn and Evans, 2010; How et al., 2017; Ravier and Buoncristiani, 2018). Four of these lakes (Nos. 4, 5b, 7, 8) are located on the same hydraulic gradient line, suggesting that if englacial conduit had not been completely closed, they share water through same englacial conduit. Whereas the other lakes (Nos. 2, 3, 5a, 10, 11, 16, 17, 18, 19, 20, 21, 22) are not matching the line. Lakes 3 and 5a are slightly above the hydraulic gradient line, suggesting that

lake 5 did not contact to main englacial conduit completely. However, after the first drainage, the lake fit the hydraulic gradient line (5b), presumably because the englacial conduit then was shared with four lakes (Nos. 4, 5, 7, 8). Although lake 1 also drained in the drainage event (Nos. 4, 5, 7, 8) in 17 July 2018, it does not lie on the hydraulic gradient line. This suggests that it drained through a different englacial conduit. Lake 1 also differs from the others (Nos. 4, 7, 8) in having a different increase rate of water-level (Fig. 12a) and in 2017 its water-level variations differed from that of the other four

lakes.

The three lakes with nearly the same increase rate in water level (Nos. 4, 7, 8) all fall on the same line in Fig. 12c. As they also discharged simultaneously, they likely shared the same main englacial conduit. If this correlation is general, then lakes with the same increase rate and lying on a hydraulic gradient line may be more likely to share the same main englacial conduit. If this hypothesis is correct, then several lakes showing this phenomenon may sometime experience simultaneous

large-scale drainage due to an opening of their main englacial conduit. If so, using satellite data to monitor the variations of area and water levels of supraglacial lakes may help reduce damage caused by large-scale drainage events.

## 6 Conclusion

We investigated the daily water-level variations of supraglacial lakes on the southern Inylchek Glacier in 2017, 2018, and

2019. We observed the simultaneous drainage of five lakes, and argued that each lake must have been connected to the same main englacial conduit via a branch englacial conduit. We also observed drainage events occurring from only one lake, and



argued that the event indicated an opening of a branch englacial conduit to the main englacial conduit. The englacial conduit system changed over a timescale of months.

The water level of the supraglacial lakes tended to increase gradually before drainage. If several lakes shared the same main englacial channel and were on the same hydraulic gradient line, then they also showed similar rates of increase in water level. Therefore, by monitoring water levels, one might be able to determine which lakes share the same englacial channel.

*Acknowledgements.* We would like to express thanks to B. Moldobekov, R. Usubaliev, A. Osmonov, E. Azisov R. Kenzhebaev of the Central-Asian Institute for Applied Geosciences (CAIAG), Ak-Say Travel, Mizunuma of NSi Co. Ltd.. Y. Mori, H. Takadama, S. Okuyama, M. Yamamoto of Niigata University. This study was supported by Grant-in-Aid for Scientific Research (B) 16H05642 and of the Ministry of Education, Culture, Sports, Science and Technology (MEXT), and the Sasakawa Scientific Research Grant from The Japan Science Society, and Inoue Field Science Research Fund from The Japanese Society of Snow and Ice.

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



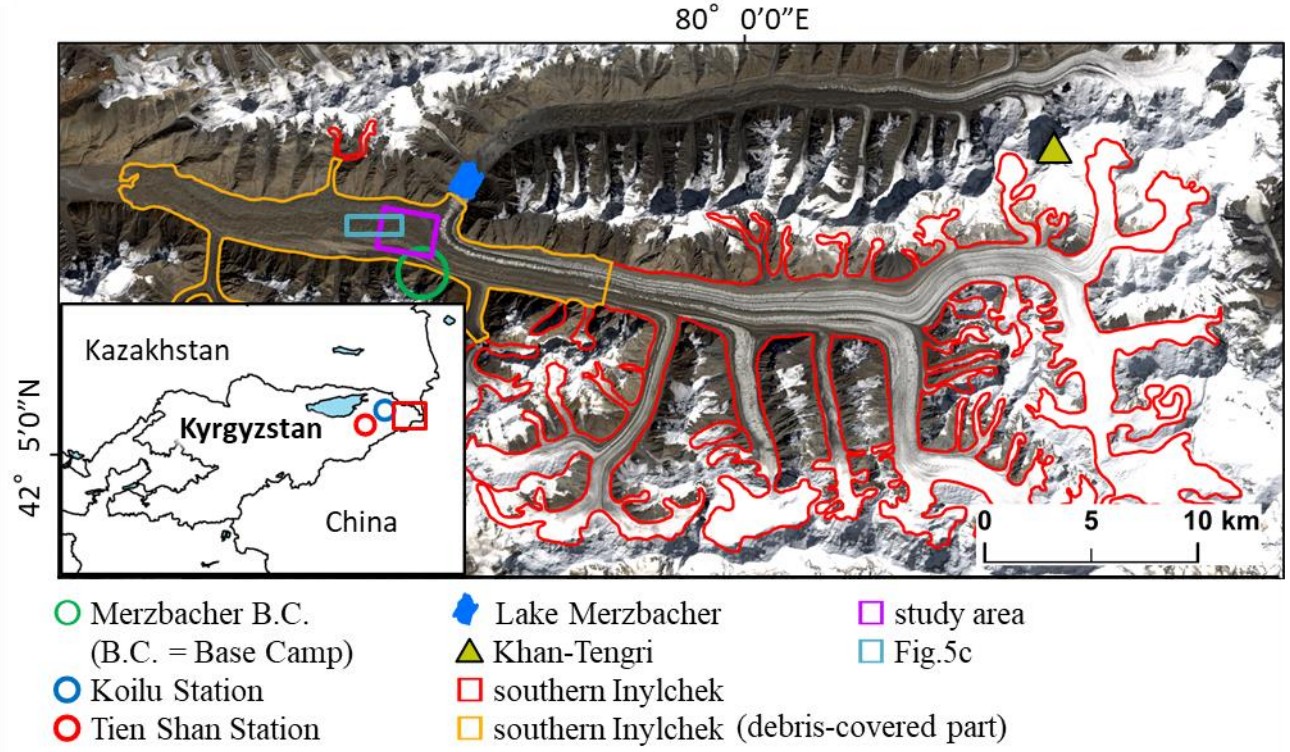

**Figure 1. Southern Inylchek Glacier in central Tien Shan, Kyrgyzstan. Red line borders the area with glacier ice, the yellow line shows the debris-covered area. The glacier outline is adapted from the GAMDAM glacier inventory (Sakai, et al., 2015). The purple box is the area shown in Fig. 2. Landsat8/OLI image taken on 4 September 2017.**



**Figure 2. Area imaged by UAV (purple box in Fig. 1). (a) The study area in 2017 (2 × 2 km²). (b) Same as (a) except 2018 and larger area (2 × 2.5 km²). (c) Same as (b) except for 2019. Red circles are GNSS survey points. Green triangles are locations with a time-lapse camera. Purple squares are locations with water-level logger. Blue areas are supraglacial lakes. Regions shown in subsequent figures outlined with colored boxes and labeled. (d) Distribution of lake sizes in the study area on the dates shown.**

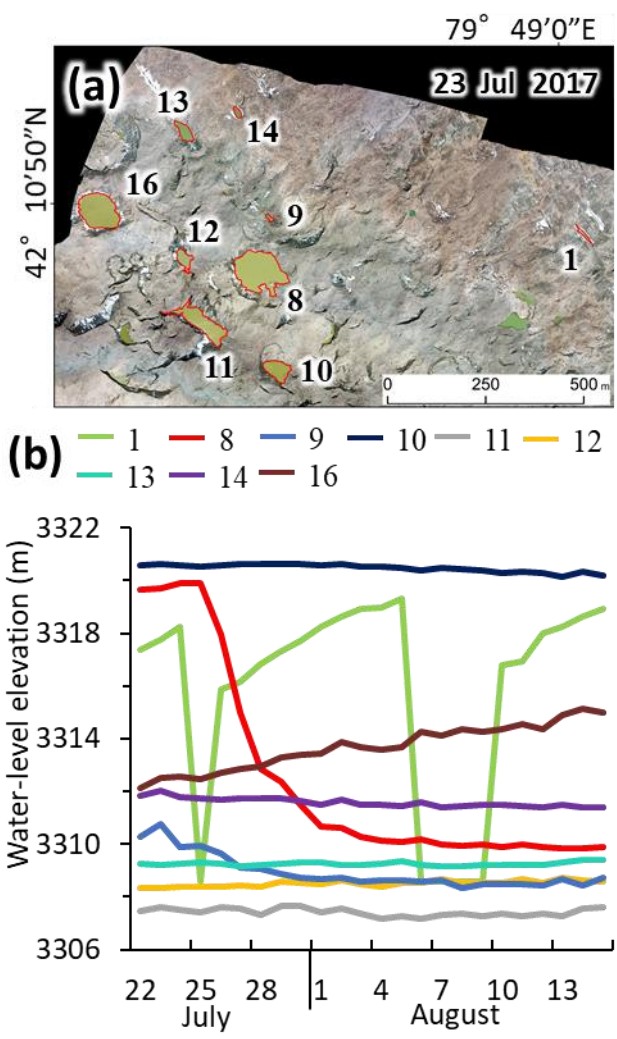

**Figure 3. Lake distribution and water-level variations from July to August 2017. (a) Location of lakes with ID numbers. Region is the yellow rectangle in Fig. 2a. The ortho-image was made from UAV data on 23 July 2017. (b) Daily water-level variations (based on DSM data) in numbered lakes in (a).**







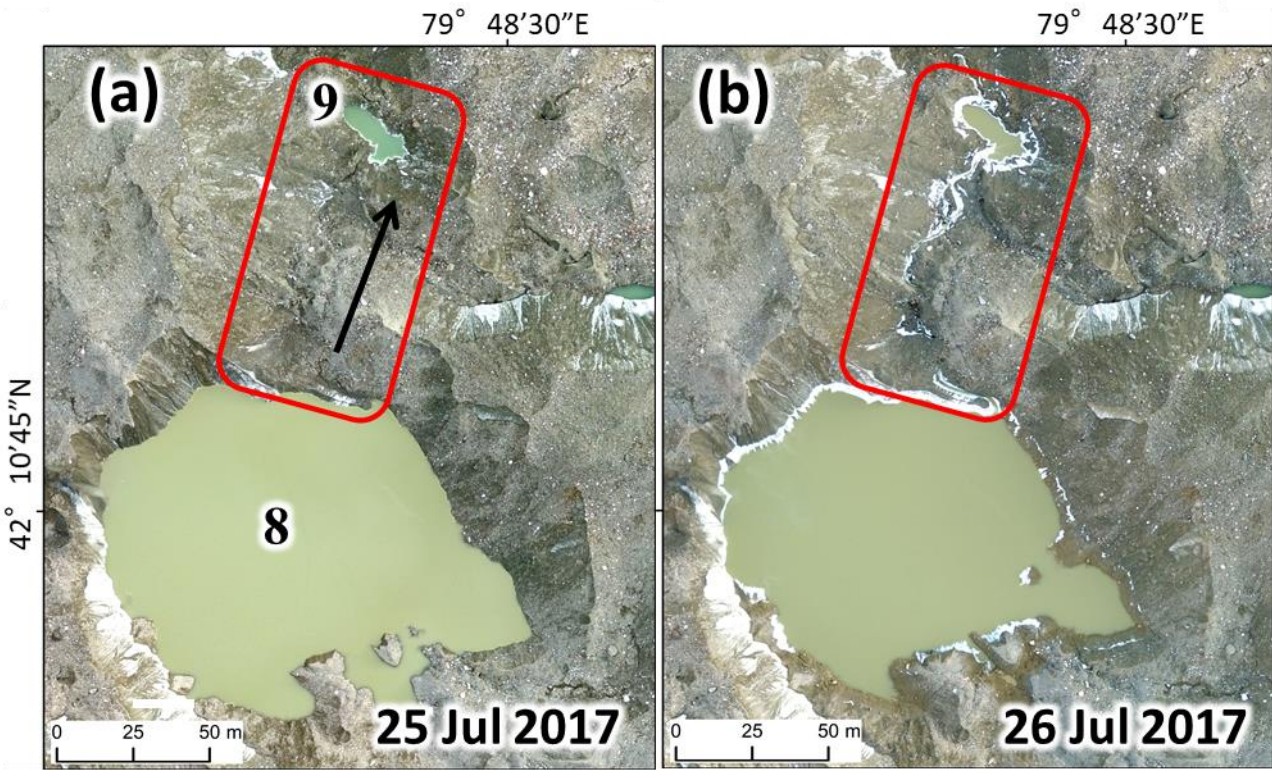

**Figure 4. UAV ortho-images showing lakes 8 and 9 before and after drainage. Red rectangle shows white region where ice wall along englacial conduit was exposed after drainage.**





**Figure 5. Drainage event on 13 May 2018. (a) Water-level variations of four lakes from water-level logger data. Lake 8 discharged at 5:00 AM on 13 May 2018 (GTM +9:00). (b) Time-lapse camera images of lake 4 that discharged on 12–13 May. (c, d) PlanetScope and Sentinel-2 images taken on 12 and 13 May. Left images (blue box) show the extended area of blue box in right images. Yellow circles show lakes of simultaneous drainage event, yellow boxes show regions of englacial conduit where most water flowed.**




**Figure 6. Lake distribution and water-level variations in July 2018. (a) Location of lakes with ID numbers on 17 July 2018. Region is the yellow rectangle in Fig. 2b. IDs with orange highlight are lakes that drained simultaneously, IDs with white highlight are lakes unaffected by the multi-lake drainage event. IDs with green highlight are lakes that drained on another day. The ortho-image was made from UAV data on 17 July 2018. (b) Same as (a) except the next day. (c) Daily water-level variations (based on DSM data) in lakes that simultaneously drained. Lake 8 includes former lake 9 of 2017 due to a merging of the lakes prior to this image. (d) Same as (c) except for the lakes that did not simultaneously drain.**







Figure 7. Field photos near several lakes. (a, b) Lake 7 before and after drainage on 17 July 2018. Red circles show landmark stones. (c) Lake 4 after drainage on 17 July 2018. (d) Lake 5 after drainage on 17 July 2018. (e) Lake 3 situation after drainage on 16 July 2018. (f) Branching in englacial conduit at location marked in Fig. 2a.







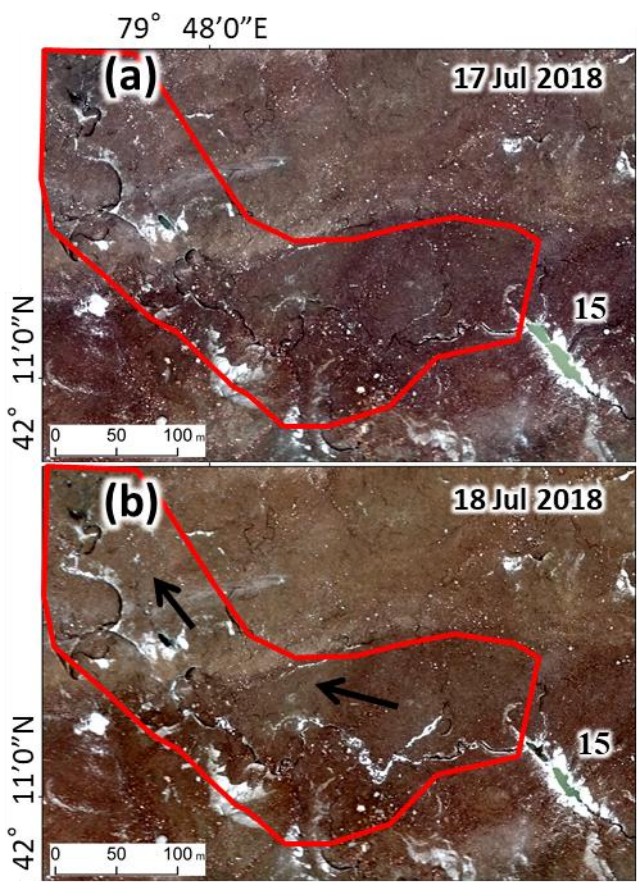

**Figure 8. Drainage of lake 15 on 17 July 2018. Drainage route is marked by black arrows in (b). New white areas in (b) show newly**
**exposed ice. Red line shows region of englacial conduit where most water flowed (black arrows). Ortho-images (a, b) were made**
**from UAV data on 17 and 18 July 2018.**





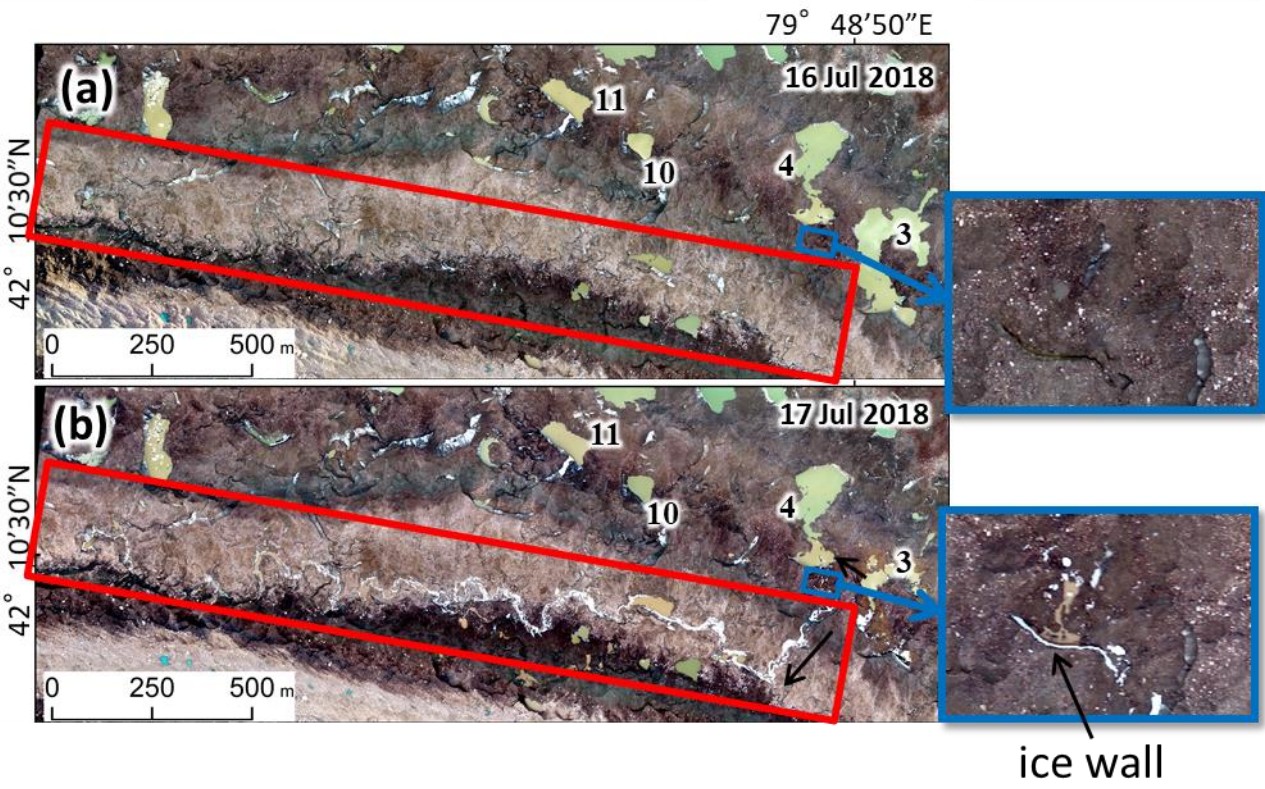

**Figure 9. Drainage of lake 3 on 16 July 2018. Newly exposed ice occurs in (b) along the englacial conduit with some parts exposed on the surface due to debris washed away during the drainage. Black arrows show water-flow directions. Some lake water flowed to lake 4. Seven lakes drained the next day. Blue box at right shows close-up of inset region. Red box shows region of englacial conduit where most water flowed. Ortho-images (a, b) were made from UAV data on 16 and 17 July 2018.**





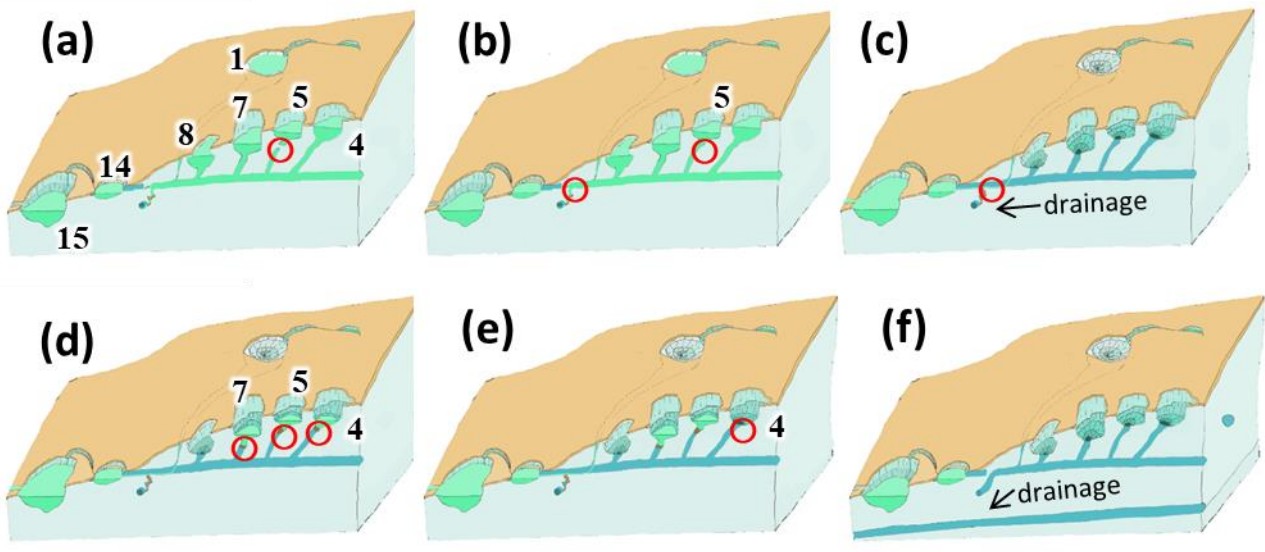

**Figure 10. Estimated englacial network changes in the study site. (a) Distribution of lakes and englacial conduits. (b) First, lake 5 drains due to a newly opened connection between its branch and main englacial conduits. (c) Simultaneous drainage of four lakes due to opening of the main englacial conduit. (d) Recharging of each lake due to closure of the branch englacial conduits by debris and ice. (e) Re-discharge of each lake due to opening of a branch englacial conduit. (f) Simultaneous drainage event in May 2018 due to the opening of another route of the main englacial conduit.**

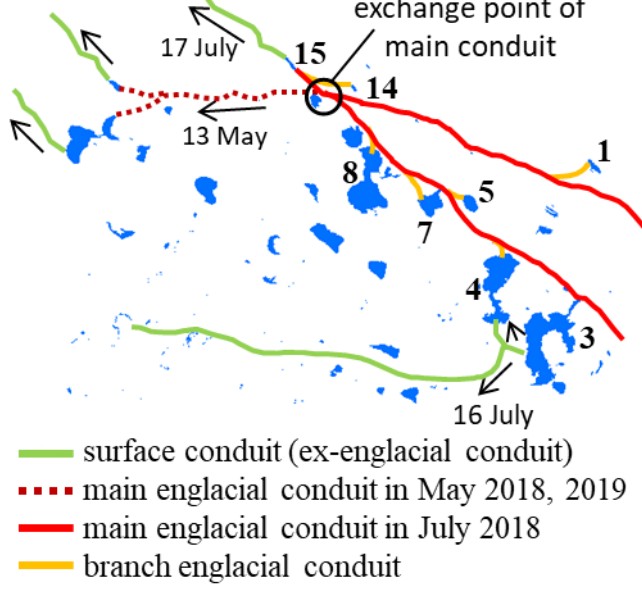

— surface conduit (ex-englacial conduit)
···· main englacial conduit in May 2018, 2019
— main englacial conduit in July 2018
— branch englacial conduit

**Figure 11. Estimated locations of the main and branch englacial conduits. Exchange point of channel shows the location where main englacial conduit changed from May 2018 to July 2018.**




**Figure 12. Characteristics of supraglacial lakes before simultaneous drainage event on 17 July 2018. (a) Water-level increases of lakes 1, 4, 7, and 8. (b) Distribution of supraglacial lakes. Orange lakes are related to the drainage event. Blue lakes are not related to the drainage event. (c) Water-level difference and distance to lake 8. Orange points are lakes related to the drainage event. Blue points are lakes not related to the drainage event. Black point is lake 5 before the first drainage on 8 July 2018.**