# Peer review of "Daily water-level variations of supraglacial lakes in the southern Inylchek Glacier, Central Asia"

_The Cryosphere, 2020_

## Referee Comment (RC1) · Anonymous Referee #1 · 28 Apr 2020

Sakuri et al. present valuable field data revealing insights into glacier hydrology for an area of supraglacial lakes on a glacier in Kyrgyzstan. The data are generally well presented, but could be more ambitiously interrogated to present additional analysis in support of the aim 'to better understand storage in and drainage through supraglacial lakes and englacial conduits'. I have suggested some further analysis to strengthen the results of the paper, and I believe the revised paper would make a valuable contribution to The Cryosphere.

General comments:

- The introduction and study site sections should be refined to better explain why the study is important and relevant, both locally and regionally. For example, the focus is on flood events from supraglacial lakes on other glaciers, but it's not clear if this is a potential hazard for your study site. Also, what is the broader status of this glacier. Give more detail when you state 'many supraglacial lakes have developed...'(L70). Is water storage increasing? Is a proglacial lake likely to develop in the future? Are the water resources important for downstream communities and infrastructure etc?
- Why are temperature data from the Hobo U20 loggers not presented? This could give further insights into the lake drainage, and the ablation associated with the water storage and drainage. There are papers in the literature looking at pond temperature and links with ablation.
- It would be useful to summarise the field data acquisition in a table, showing the number of surveys, number of images for each survey, number of ground control points.
- The volume time series of the lakes are known from the DEM and water level data but are hardly discussed. It would be useful to present the volume changes of each lake, in addition to the water level elevation change (which relates to a different volume increase for each lake). There are also empirical area-volume relationships presented by Cook and Quincey (2015) and more recent papers, which your data could make a valuable contribution to. Presenting time series volumetric change based on the known bathymetry of the pond and the water level could reveal more insights into lake drainage.
- It would be useful to present DEM differences of the study area to reveal topographic change associated with the drainage events e.g. Thompson et al. (2016) and Miles et al. (2018), to look for evidence of topographic change associated with englacial conduits, and to look into evidence of connectivity between the ponds (e.g. your Figure 12).

Specific comments:

L33. Note that in at least one of these examples (Rounce), englacially storage water was also suggested as a source.

L40. Englacial or supraglacial conduit.

L51-53. This paper mainly dealt with longer term changes in supraglacial water. Watson et al. (2017) deal with the seasonal expansion/drainage and thermal regime of ponds.

L87. Specify the number of ground control points in each year. What features were used for the GCPs?

L88. It is not clear what this accuracy refers to. The absolute position accuracy following post-processing, or is it referring to your models.

L89. Provide more detail about the water level setup. What is the stated accuracy and what barometric data were they processed against?

L92. Specify details about the UAV, including make, model, camera specifications and parameters, photo interval.

L94. Clarify this sentence. Do you mean most of the workflow was automatic, but that you manually added the GCPs? Were the location of the images from the UAV used in the processing?

L96. How was the 'standard DSM' chosen? Based on quality?

L98. I assume this is an XY shift in ArcGIS, but what about the Z shift due to the slope of the glacier. What is the slope of the glacier in the survey location and therefore do the ponds lower in elevation as they move downglacier?

Is this the flow speed for your study site, or an average for the whole glacier?

L105. Landset-8. Check throughout.

L114. It is not clear if these values represent some sort of average or range. You should report in a table, all water levels from the UAV models and the corresponding water level logger measurements, i.e. for all days.

L133. It would also be useful to compare with studies that have used empirical relationships for estimating supraglacial lake area-volume e.g. Cook and Quincey (2015), Watson et al. (2017). This could give you a better indication about the overall water storage for all lakes on the glacier, how much your site contributes to the total water storage, and how much seasonal drainage volume there is.

L189. 'Sentinel'.

L199. 'Connection'

L269. '...different increase rate of water-level'. The rates are not directly comparable without considering the different pond bathymetries, i.e. presenting a volume time series.

L270. Quantify the rate of water level increase.

L276. Is there potential for damage caused by drainage from this glacier? It should be mentioned in the introduction or study area section.

L380. Provide more lat/lon grid markers around the edge of figures (check throughout). It seems the distinction between debris-covered and clean ice is an arbitrary abrupt line, rather than including medial moraines etc. I'm not sure this distinction is helpful or necessary.

L485. Add a scale to this figure.

L492. Clarify what you mean by 'water level difference'. Is this relating to the water level change in those lakes compared to lake 8?

References

Cook, S.J. and Quincey, D.J. 2015. Estimating the volume of Alpine glacial lakes. *Earth Surf. Dynam.* **3**, pp.559-575.

Miles, E.S. Watson, C.S. Brun, F. Berthier, E. Esteves, M. Quincey, D.J. Miles, K.E. Hubbard, B. and Wagnon, P. 2018. Glacial and geomorphic effects of a supraglacial lake drainage and outburst event, Everest region, Nepal Himalaya. *The Cryosphere.* **12**(12), pp.3891-3905.

Thompson, S. Benn, D. Mertes, J. and Luckman, A. 2016. Stagnation and mass loss on a Himalayan debris-covered glacier: processes, patterns and rates. *Journal of Glaciology.* **62**(233), pp.467-485.

Watson, C.S. Quincey, D.J. Carrivick, J.L. Smith, M.W. Rowan, A.V. and Richardson, R. 2017. Heterogeneous water storage and thermal regime of supraglacial ponds on debris-covered glaciers. *Earth Surface Processes and Landforms.* pp.229-241.

---

## Referee Comment (RC2) · Sam Herreid (Referee) · 4 May 2020

The article titled "Daily water-level variations of supraglacial lakes in the southern In-ylchek Glacier, Central Asia" by Naoki Sakurai et al. uses an impressive dataset of daily high-resolution DSMs acquired from repeat drone flights over a subset of In-ylchek Glacier for a window of time repeated over three consecutive years. Using these data, the authors record and interpret several supraglacial lake drainage events including instances where several lakes drain simultaneously. The authors go on to infer the location and behavior of the englacial drainage system during this time. I en-joyed reading this paper and found it mostly easy to follow. My comments below are

[Figure]

mostly minor, and I think a revised version of this manuscript would be well suited for publication in The Cryosphere. Some of the main points include the following: unclear langue regarding englacial and supraglacial flow, which was confusing throughout the manuscript. A horizontal correction was applied to the repeat DSMs, but a discussion of, or a correction for, vertical motion from factors like emergence, elevation loss due to moving down a slope and ablation was never addressed. There was a 45% filter to the lake elevation data which seemed high to me, I would like to better understand why so much of the lake elevation data seems to have had a non-zero surface slope. Lake volume changes were measured but only sparsely reported as context, I think this paper would benefit from having these results presented more completely. The paper does a good job providing a detailed chronology of events for this portion of this one glacier but most readers won't necessarily care about this particular location and will be looking for takeaway points that further our general process understanding of the system, which are thus far largely missing. These points, and additional minor comments, are detailed below.

L32-33: "In none of these cases did researches find evidence for a large proglacial lake prior to the drainage, but in all cases large supraglacial lakes had drained" Is there an unstated assumption here that we would only see drainage events from glaciers with a lake in front of the terminus? I'm not aware of this assumption, so maybe cite some occurrences of this in the literature.

L38-39: "Thus, recent large-scale drainages are related to supraglacial lakes and englacial conduits on debris-covered glaciers." This is a little odd with "recent" suggesting that these physics were different in the past? Better might be "These recent studies establish a coupling between large-scale drainage events and supraglacial and englacial water storage dynamics."

L41: Can you say how lake development is dependent on flow velocity? In principal a lake could grow on a flat portion of glacier moving at any speed. What is the intermediate process that prevents or allows lake formation? A flow velocity gradient coupled

with crevassing? A feedback loop with debris cover? Or a more efficient englacial channel system?

L43: This reference is recent and relevant to the acceleration of glacier ablation with respect to lakes/ponds: Miles, Evan S., et al. "Surface pond energy absorption across four Himalayan glaciers accounts for 1/8 of total catchment ice loss." Geophysical research letters 45.19 (2018): 10-464.

L46: This is written as, but does not seem to be, an exhaustive list of the mechanisms for englacial conduits closure and reopening. Thermal erosion comes to mind. Also similar to above, how exactly does glacier flow cause englacial conduits to open and reopen, there are some missing steps, I think. In the two paragraph begin at L49 you describe well supported seasonal evolution of lake systems. Can you also cite some long-term trends for context?

L57: Change to: "The timing of the maximum number.."

L60: Maybe this is a useful citation here as well: Bartholomaus, Timothy C., Robert S. Anderson, and Suzanne P. Anderson. "Growth and collapse of the distributed subglacial hydrologic system of Kennicott Glacier, Alaska, USA, and its effects on basal motion." Journal of Glaciology 57.206 (2011): 985-1002.

L63: Maybe say "[primarily] using an unmanned aerial vehicle" since you also use satellite data.

L65: Probably cite Fig. 1 here.

L68-69: "with surface-flow velocities in its upper part being faster than those in its lower part" This does not relay much information to the reader. Are your upper and lower in reference to where there is more debris and less debris, or do you mean this glacier has a particularly fast flowing accumulation zone? Consider taking this one step further so we know how this is useful to know.

L71: This annual seasonal drainage cycle from Narama etal., 2017 might be worth

describing here for another sentence since this seasonal evolution sounds particularly relevant to the rest of this study.

L72-77: Are meteorological data used beyond the statistics reported here? I don't think it's needed to establish abbreviated names and might not need to be in Fig. 1.

L82: Lake Merzbacher base camp is referenced like it is established or somewhere we should know. I assume it wasn't established just for this study?

L82: This is very nice that you were able to fly every day. Can you please comment if you flew in rain, strong wind and fog, or if you adjusted your flight times to accommodate bad weather? Or maybe there was none?

L86: "...as this period was hardly affected by weather and sunlight." By "affected by sunlight" I assume you mean it was high enough above the horizon to illuminate the surface but not overhead to cause cast shadows. If this is correct could you state it, or be more precise in what you mean. Also, in the meteorological paragraph above maybe you could say that it's common to have good weather early in the morning during this time of year, or some similar support to your saying here that the weather is almost always good.

L88: Do you mean to say you obtained an accuracy of around 20 cm? >20cm could mean almost anything.

L92: Do you have any constraint on the z accuracy of your DSMs? Did you find some to be better than others in your stack, and if so, any variable that you suspect is responsible?

L93: This is a strange location for a reference to Fig 2. I would remove.

L94: The sentence "Workflow of SfM software was almost automatic without setting of GCPs which we measured." Is unclear and needs to be explained further.

L96: [an] ortho-image and DSM [remove: data] as the...

L96: This first step is unclear, are you shifting the DSM in only x,y space or also in some averaged z? There are competing signals of emergence, glacier melt and elevation loss as the features move down slope. With some more specific information about this glacier you may be able to make an argument to neglect these terms, e.g. your measured dz is much greater than the sum of these terms over the observation period or that the observation spatial domain is small enough that emergence at least can be assumed uniform, but they need to be addressed in some way since you compare absolute elevation data. It seems logical you would use the first image in a stack as the reference image but that isn't stated here, why not?

L99: Was the shift to correct for glacier flow linear? Meaning did you use only one tie point or several that then caused a nonlinear translation?

L101, Step 4: Are these points throughout the polygon or only along the perimeter?

L103: Excluding 45% sound like a lot. How did you choose this value? You must know a lot about SfM errors over the flat surface of water, could you add some detail about the errors? An oblique angle image of the ortho-image draped over the corresponding DSM could be useful for the reader to see and understand the source data and why a 45% filter was applied.

L103-104: You mention methods here for changes in water volume and periodically report values in the text but these results are never presented on their own. I think there is a lot of potential to tell a more complete story by presenting these findings as well.

L105-107: Lake area variations are not shown in any table or figure, I can see this being a useful measure to bridge the unknown periods between your windows of surface elevation measurements but this is not shown in your results, nor a discussion of how lake area and dz are coupled.

L111-114: All field glaciologist will empathize with deployed sensors not returning useful data, but I don't think it's necessary to describe here. If the sensors in lakes 4 and 10 did not record any useful data, then it would ease the reading of this section to remove them from this paragraph altogether.

L125: Here you say the larger lakes are in the central part of the glacier, I assume you mean orthogonal to flow, not along flow, but this is not clear.

L127: I like the organization and discussion of drainage events, however I think there are still some bulk results that you can make here. Specifically, more comprehensive results of volume changes and maybe analysis of the spatial distribution of draining and not draining ponds as you show in the figures with colored numbers.

L129: "Consider the storage and drainage behavior of the nine lakes marked in Fig. 3a" Is an unnecessary sentence. Consider starting this section with a statement.

L131 and throughout: "...increased gradually by 0.3 m (2723 m^3)" Is this a rate per day? Please specify the time here. And considering sidewall slope, the volume will be different for identical surface changes or losses. Consider reporting an integrated change rather than a rate or be clear with assumptions/averages.

L136: Did you measure lake area change? It would be interesting to see a plot of lake area change vs volume change and I believe you have all the data to plot this. A statistically significant relation would make some inference on lake ice wall geometry.

L136: "a conduit with exposed parts" this is a recurring concept throughout this article that seems confusing to me. In my mind a conduit in glaciology is an englacial channel, while surface streams can thermally erode channels and sometimes briefly go under ice bridges, but I wouldn't call them a conduit. It's fine that you use terms differently, and perhaps my view is incorrect, but maybe in the introduction you could write a short section or few sentences describing how you define a conduit.

L137: "[That exposure parts, change to: These ice exposed parts] were caused by erosion from the drainage water..." I assume you mean erosion of both ice and debris

cover, if so, please state that to be more clear.

142-143: You write that lake 1 had a gradual increase of water, but in the sentence before you said it rose by 7.5 m in one day before the next drainage. This seems like a rapid refill to me, what do you mean exactly by gradual?

L145: For the lakes that maintain a stable water level do you have evidence, e.g. from satellite data, that they are also draining just not during this period? Can you say if any lakes persist throughout the course of a year or more?

L146: This section is essentially a figure caption. I think you should bring out more of the significance of why this event is one of your key results or consider merging it to a more general section listing events and citing figures. One key result that seems missing from this event is a plot of logger vs DSM derived change in water surface elevation.

L165: rewrite: increased by 0.6-2.0 mˆ-1

L167: For the lake(s) that gradually filled and then maintained a stable level, what can you infer about the water table of the glacier at this location and the bulk permeability?

L169: It's great that you use satellite imagery to complement your timeseries but it leaves us guessing when you were monitoring this region. Could you please add a timeline figure or subfigure that shows the sensor, datatype and timing of each acquisition you used for this analysis.

L172: Change to "This result suggests the drainage route was…" and later "lakes 14 and 15 after the 13 May.."

L174: "For [the] drainage.."

L174: That the "conduit" is visible in Fig. 8 means it is a supraglacial stream or channel not englacial.

L176: The sentence starting "The drainage route with exposed ice wall…" Should be

rewritten more clearly, I think your key point is the water took a different drainage path but the positioning relative to ice walls is unclear.

L177: Consider: "This difference indicates a change in drainage route between"

L182: Cite a figure and consider just describing the outcomes here that are notable, it currently reads more like a literal description of a figure.

L189: This sentence would also benefit from citing a timeline figure of data sources. Maybe above the timeline you could have the number of the lake that drained at that time.

L195: Why was the route unclear? There was no geomorphological change evident? Does this suggest it was a smaller event? Or are you considering the drainage went truly englacial?

L201: "..despite being." This sentence is incomplete.

L204: "indicating that some part of the englacial conduit system changed" I would like to have a sense for what is truly englacial and what is supraglacial.

L205-207: This is almost a direct quite from the introduction.

L207-208: How can you be sure there wasn't always some flow through the lake? In other words, is the water truly cut of during this time, as you state here, or is there a, possibly small, but constant in and out flow while the lake level is stable?

L216: give the figure number along with the letter for each reference, here and below.

L217 Do you have any evidence suggesting why a lake would partially drain? The conduit is not at the deepest point of the lake? Or there is a blockage during the first half of draining?

L239: You can't really say "this observed behavior is consistent with the distribution of englacial conduits shown in Fig. 11" because you drew the conduits shown in

Fig. 11. You can say "We estimated the location of conduits following the theoretical model put forth by x (citation) and guided by the behavior we observed" Or were these supraglacial channels that you could map and not infer?

L243-245: Does conduit reuse have a limiting depth within the glacier? I assume at some depth overburden pressure will close any unpressurized channel.

L251-254: The sentence beginning " This behavior indicates that the opening" is unclear. It starts as a general statement "of an englacial conduit" but then changes to a past tense description of a particular event "maybe have been caused by." The further point relating water pressure and water volume is also not clear to me. The following sentence is also unclear, I believe the conduit would need to be pressurized to have commutative lake level fluctuations.

L260-261: Please explain further how other studies found use for hydraulic potential and if they also found a linear relation with a similar slope. A gradient from 5 points is not very convincing, could you please find/plot this gradient for all of the multi-lake drainage events you constrain and then build on your interpretation of what this means or infers.

L275-276: What measure would be needed between monitoring from satellites and reducing damage? I think a key first step is establishing how robust these channel patterns are over many years.

L282-283: It's not enough to say "The englacial conduit system changes over a timescale of months", you need to qualify this with a summary of what exactly changed.

L286: Why not take this one step further and consider lake area change to infer conduit connectivity?

Figure 1: I don't understand breaking the glacier into a debris covered part and the rest of the glacier along an arbitrary line, roughly mapping the debris cover for this figure would be more logical.

I don't see the reason for putting Koilu Station and Tien Shan Station on this map or in this paper.

Figure 2: 3 GNSS points concentrated in the lower right of panel C doesn't seem like very many ground control points. Was there a loss of DSM accuracy with increasing distance from these ground control?

Why did you change the lake color to blue? The red lines show which lakes you identified and any variation in turbidity captured in the color of the water itself is also interesting.

Figure 3 (and Fig 6): I like these figures a lot, however it would be slightly easier to follow if rather than random colors, the lakes and water-level elevation lines were colored in a gradient color-scheme with distance from the lowest down glacier point in the image. This would allow the reader to easily know the relative elevation ranking of each lake.

Figure 3: Why was the obvious lake in the lower right excluded?

Figure 4, 8 and 9: If you can see the expression of ice and rock erosion of a drainage event from a drone in the sky, it is be definition not englacial. Please correct this throughout the manuscript. There may be instances where the water is englacial, particularly at the lake itself, but I think a clear distinction needs to be made.

Figure 4: Since this lake does not drain completely and looking at the drainage path, it looks more like a thermally eroded ice dam that caused the partial drainage.

Figure 5: Please add here or in a separate figure the same water level variations from both the logger data and the DSM data.

Figure 10: After considering the semantic, englacial vs supraglacial channel points made several times above, please re-evaluate this figure and insure it appropriately shows water below the surface only when there was in fact water flowing in channels below the surface.

Figure 11: Missing scale. It also might help to have a vector showing the general direction of ice flow.

[Figure]

---

## Author Comment (AC1) · 10 Jun 2020

First of all, we would like to thank the referees for their constructive and detailed comments which helped to improve the manuscript. We have tried to incorporate for reviewer comments.

Although both reviewers pointed out about area and volume changes of supraglacial lake, we already checked water balance between outcome and income at simultaneous drainage event before. However, we did not get good balance. As the reason, water volume of englacial conduit might be larger, or the connecting network of supraglacial lakes might be wider area (at least 1km2). Thus, water volume is needed under more consideration. We try to analyze this subject more for next paper. We also can prepare for two supplement files about timeline of event and satellite data, and water level data from DSM and datalogger.

Blue text is comment of reviewers, and black text is answer from authors.

**Reviewer1:**

1. The introduction and study site sections should be refined to better explain why the study is important and relevant, both locally and regionally. For example, the focus is on flood events from supraglacial lakes on other glaciers, but it's not clear if this is a potential hazard for your study site. Also, what is the broader status of this glacier. Give more detail when you state 'many supraglacial lakes have developed…'(L70). Is water storage increasing? Is a proglacial lake likely to develop in the future? Are the water resources important for downstream communities and infrastructure etc?

   A: The large-scale drainage and flooding from the debris-covered glacier is not regional issue, and this phenomenon can occur in any debris-covered glaciers. When considering large drainage, the understanding of phenomena of supraglacial lakes including water storage and drainage is not enough. In this study, we tried to clarify a part of the phenomena based on field work and satellite data analysis. Daily fluctuations of the lake phenomena are not reported yet. In the study site, we do not know that the large-scale drainage and flooding occurs or not. The proglacial lake dose not develop because terminal moraine is absent. However, these are not important, we focused on the phenomena of water storage and drainage of supraglacial lakes which might be related to large-scale drainage from debris-covered glacier without proglacial lake.

   We added the sentence as "The large-scale drainage and flooding from the debris-covered glacier is not regional issue, and the phenomenon can occur in any debris-covered glaciers. To better understand the storage in and drainage through supraglacial lakes and englacial conduits which might be related to large drainage,,," and we changed "developed" to "distributed".

2. Why are temperature data from the Hobo U20 loggers not presented? This could give further insights into the lake drainage, and the ablation associated with the water storage and drainage. There are papers in the literature looking at pond temperature and links with ablation.

   A: We checked the relationship between water temperature and lake variation. Although water variations become small range (around 1-2 °C) inside of water, we do not get the special evidence before drainage. We showed the temperature data in new Figure 5a.

   We changed the sentence as "Figure 5a shows water-level and temperature variations based on water-level data loggers…."

3. It would be useful to summarise the field data acquisition in a table, showing the number of surveys, number of images for each survey, number of ground control points.

A: Although we can show the table as a supplemental data, we already wrote these data on the manuscript.

4. The volume time series of the lakes are known from the DEM and water level data but are hardly discussed. It would be useful to present the volume changes of each lake, in addition to the water level elevation change (which relates to a different volume increase for each lake). There are also empirical area-volume relationships presented by Cook and Quincey (2015) and more recent papers, which your data could make a valuable contribution to. Presenting time series volumetric change based on the known bathymetry of the pond and the water level could reveal more insights into lake drainage.

   A: We already investigated the water balance between inflow and outflow (drainage) based on water volume in detail. However, water balance is not coincided each other. As the reason, volume of englacial network might be larger or the connecting network of supraglacial lakes might be wider area. Thus, water volume is needed under more consideration. We try to analyze more for next paper.

5. It would be useful to present DEM differences of the study area to reveal topographic change associated with the drainage events e.g. Thompson et al. (2016) and Miles et al. (2018), to look for evidence of topographic change associated with englacial conduits, and to look into evidence of connectivity between the ponds (e.g. your Figure 12).

   A: Although we also confirmed the differences using DSMs, we could not find large topographic changes before and after drainage.

●Specific comments

1. L33. Note that in at least one of these examples (Rounce), englacially storage water was also suggested as a source.

   A: We changed the sentence as "Khumbu region of eastern Nepal had large-scale drainage from a large supraglacial lake and englacially storage water (Rounce et al., 2017)."

2. L40. Englacial or supraglacial conduit.

   A: We changed the sentence as "englacial or supraglacial conduit"

3. L51-53. This paper mainly dealt with longer term changes in supraglacial water. Watson et al. (2017) deal with the seasonal expansion/drainage and thermal regime of ponds.

   A: We changed the sentence as "We distinguished the chapter of annual and seasonal variations including Watson et al., 2016; 2017."

4. L87. Specify the number of ground control points in each year. What features were used for the GCPs?

   A: We used GCPs to make ortho-image and DSM. The DSM was used to correct relative water level fluctuations to elevation values.

5. L88. It is not clear what this accuracy refers to. The absolute position accuracy following post-processing, or is it referring to your models.

A: We changed the sentence as "We obtained an absolute position within 30 cm accuracy at GCPs positions by post-processing with data from Kyrgyz GNSS reference station."

6. L89. Provide more detail about the water level setup. What is the stated accuracy and what barometric data were they processed against?

A: We used two water level data loggers on lake bottom and land. Water logger data at lake bottom was corrected by atmosphere pressure. The accuracy of water-level logger sensor (Hobo U20) is ±1.5 cm. We compared water level from DSM with water level from data logger.

We changed the sentence as "We also set water-level data loggers (Hobo U20) at water and atmosphere on ground with an interval of 1 hour, and time-lapse cameras (Brinto) with an interval of 1 day from 2016 to 2019 (Fig. 2)." And "Water-level data taken from UAV ortho images was tested against direct measurements from water-level loggers corrected by atmospheric data logger. The accuracy of water-level logger corrected by atmospheric data were ±1.5 cm."

7. L92. Specify details about the UAV, including make, model, camera specifications and parameters, photo interval.

A: We changed the sentence as "Aerial photographs were taken using a UAV (Phantom-3 Advance and Phantom-4 (DJI): camera pixels 12.4MP, lens FOV94°, 20mm, f/2.8, ISO 100-1600) …."

8. L94. Clarify this sentence. Do you mean most of the workflow was automatic, but that you manually added the GCPs? Were the location of the images from the UAV used in the processing?

A: We run the processing automatically using the location of the images from the UAV. But we added the GCPs taken by GNSS survey manually to fix the correct location and elevation during the processing. We changed the sentence as "Although workflow of SfM software was almost conducted automatically using the location of the images from the UAV, we set GCPs manually taken by Trimble GeoExplorer6000 in GCP setting."

9. L96. How was the 'standard DSM' chosen? Based on quality?

A: We decided that the DSM when each lake is the smallest is the standard DSM. Thus, the standard DSM is different for each lake. We changed the sentence as "We set an ortho-image and DSM as the standard for each lake when each lake is smallest during survey period. Thus, standard data is different for each lake. For example, we set ortho and DSM on 25 July as standard data for (No. lake 1) in 2017."

10. L98. I assume this is an XY shift in ArcGIS, but what about the Z shift due to the slope of the glacier. What is the slope of the glacier in the survey location and therefore do the ponds lower in elevation as they move downglacier?
Is this the flow speed for your study site, or an average for the whole glacier?

A: The surface gradient is quite gentle in the study site. Since the glacier surface flows at a speed of about 4 m/month in the study site, all other ortho-images and DSM data were fitted to the standard data by shifting them with the use of the 'georeference' function (none liner) with more than 10 tie points in ArcGIS. We just shifted the DSM in x and y space. The vertical differences are quite small for one month due to glacier motion and quite small surface gradient in the study site (2 x 2.5 km).

We changed the sentence as "Since the glacier surface flows at a speed of about 4 m/month in the study site, all other ortho-images and DSM data were fitted to the standard data by shifting them with the use of the 'georeference' function (none liner) with more than 10 tie points in ArcGIS. We just shifted the DSM in x and y space. The vertical differences are quite small for one month due to glacier motion and quite small surface gradient in the study site (2 x 2.5 km)."

11. L105. Landset-8. Check throughout.

A: We changed all.

12. L114. It is not clear if these values represent some sort of average or range. You should report in a table, all water levels from the UAV models and the corresponding water level logger measurements, i.e. for all days.

A: We removed one sentence "L128: The average difference of five lakes was −7.6 ± 8.4 cm." We added the supplement table. We added the sentence as "The water level data from DSM and data logger were shown in supplement table 1."

13. L133. It would also be useful to compare with studies that have used empirical relationships for estimating supraglacial lake area-volume e.g. Cook and Quincey (2015), Watson et al. (2017). This could give you a better indication about the overall water storage for all lakes on the glacier, how much your site contributes to the total water storage, and how much seasonal drainage volume there is.

A: We already answered question 4.

14. L189. 'Sentinel'.

A: We changed it.

15. L199. 'Connection'

A: We changed it.

16. L269. '…different increase rate of water-level'. The rates are not directly comparable without considering the different pond bathymetries, i.e. presenting a volume time series.

A: If each lakes is independent (no connection), we should consider shape of lake basin (i.e. a volume time series.). However, those lakes (Nos, 1, 4, 7, 8) were connected some englacial conduits each other, because those lakes drained at the same timing twice in a year. In this condition, increase of water level is affected by the conduit condition (for example: connection). For that reason, we here compared with the increase of water level directly.

17. L270. Quantify the rate of water level increase.

A: We showed the total changes of water level. We changed the sentence as "Total increases of water-level for each lakes (Nos. 1, 4, 7, 8) from 8 to 17 July 2018 are 1.96 m, 0.74 m, 0.58 m, 1.13 m, respectively."

18. L276. Is there potential for damage caused by drainage from this glacier? It should be mentioned in the introduction or study area section.

    A: We answered question 1.

19. L380. Provide more lat/lon grid markers around the edge of figures (check throughout). It seems the distinction between debris-covered and clean ice is an arbitrary abrupt line, rather than including medial moraines etc. I'm not sure this distinction is helpful or necessary.

    A: We added two grid lines of lat/lon, and deleted lines of medial moraines on the debris-covered area.

20. L485. Add a scale to this figure.

    A: We added the scale on the figure.

21. L492. Clarify what you mean by 'water level difference'. Is this relating to the water level change in those lakes compared to lake 8?

    A: This is relative changes from lake 8. We changed the sentence as "Relationship between distance and water-level differences for each lake comparing to lake 8."

**Reviewer 2:**

1. L32-33: "In none of these cases did researches find evidence for a large proglacial lake prior to the drainage, but in all cases large supraglacial lakes had drained" Is there an unstated assumption here that we would only see drainage events from glaciers with a lake in front of the terminus? I'm not aware of this assumption, so maybe cite some occurrences of this in the literature.

    A: We changed the sentence more clearly as "In these cases, a large proglacial lake was not reported in glacier terminus as an evidence for prior to the drainage, but in all cases large supraglacial lakes had drained."

2. L38-39: "Thus, recent large-scale drainages are related to supraglacial lakes and englacial conduits on debris-covered glaciers." This is a little odd with "recent" suggesting that these physics were different in the past? Better might be "These recent studies establish a coupling between large-scale drainage events and supraglacial and englacial water storage dynamics."

    A: We changed the sentence as "These recent studies establish a coupling between large-scale drainage events and supraglacial and englacial water storage dynamics."

3. L41: Can you say how lake development is dependent on flow velocity? In principal a lake could grow on a flat portion of glacier moving at any speed. What is the intermediate process that prevents or allows lake formation? A flow velocity gradient coupled with crevassing? A feedback loop with debris cover? Or a more efficient englacial channel system?

A: Previous studies pointed out that the development of supraglacial lakes is related to surface gradient and surface flow. We also think that englacial conduit is related to form supraglacial lake based on this research.

4. L43: This reference is recent and relevant to the acceleration of glacier ablation with respect to lakes/ponds: Miles, Evan S., et al. "Surface pond energy absorption across four Himalayan glaciers accounts for 1/8 of total catchment ice loss." Geophysical research letters 45.19 (2018): 10-464.

A: We added the reference and text.

5. L46: This is written as, but does not seem to be, an exhaustive list of the mechanisms for englacial conduits closure and reopening. Thermal erosion comes to mind. Also similar to above, how exactly does glacier flow cause englacial conduits to open and reopen, there are some missing steps, I think. In the two paragraph begin at L49 you describe well supported seasonal evolution of lake systems. Can you also cite some long-term trends for context?

A: We added thermal erosion with related reference. We also distinguished the chapter of inter-annual and seasonal variations including other references. We changed the sentence as " The appearing and vanishing of supraglacial lakes can occur when englacial conduits close and reopen, and such activity occurs due to closing of englacial channel by roof collapses, creep closure, freezing of stored water, deposition of ice and debris, and thermal erosion (Sakai et al., 2000; Gully and Benn, 2009; Narama et al., 2018). " and " We made a paragraph of the long-term trend of lake variations."

6. L57: Change to: "The timing of the maximum number.."

A: We changed it.

7. L60: Maybe this is a useful citation here as well: Bartholomaus, Timothy C., Robert S. Anderson, and Suzanne P. Anderson. "Growth and collapse of the distributed subglacial hydrologic system of Kennicott Glacier, Alaska, USA, and its effects on basal motion." Journal of Glaciology 57.206 (2011): 985-1002.

A: We added the this reference and text.

8. L63: Maybe say "[primarily] using an unmanned aerial vehicle" since you also use satellite data.

A: We added it.

9. L65: Probably cite Fig. 1 here.

A: We added it.

10. L68-69: "with surface-flow velocities in its upper part being faster than those in its lower part" This does not relay much information to the reader. Are your upper and lower in reference to where there is more debris and less debris, or do you mean this glacier has a particularly fast

flowing accumulation zone? Consider taking this one step further so we know how this is useful to know.

A: We changed the sentence more clearly as " …..surface-flow velocities in its upstream from Lake Merzbacher, being faster than those on its downstream from Lake Merzbacher (Nobakht et al., 2014; Shangguan et al., 2015, Fig.1)."

11. L71: This annual seasonal drainage cycle from Narama et al., 2017 might be worth describing here for another sentence since this seasonal evolution sounds particularly relevant to the rest of this study.

A: We already explained for the content of Narama et al. (2017) in INTRODUCTION.

12. L72-77: Are meteorological data used beyond the statistics reported here? I don't think it's needed to establish abbreviated names and might not need to be in Fig. 1.

A: We used these climatic data for more than 30 years when these were recorded in USSR. We changed to normal name of the stations. We need annual precipitation and temperature as a basic data of climate environment, because some scientists insist on the increase of precipitation as the reason of lake appearing.

13. L82: Lake Merzbacher base camp is referenced like it is established or somewhere we should know. I assume it wasn't established just for this study?

A: We added the information as "Our field survey ran during the summer in 2017, 2018, and 2019 in the middle part of the southern Inylchek Glacier near the Lake Merzbacher base camp which built by CAIAG and GFZ (Fig. 1). "

14. L82: This is very nice that you were able to fly every day. Can you please comment if you flew in rain, strong wind and fog, or if you adjusted your flight times to accommodate bad weather? Or maybe there was none?

A: We added some sentences as "However, we changed flight time on several days due to climate condition."

15. L86: ": : :as this period was hardly affected by weather and sunlight." By "affected by sunlight" I assume you mean it was high enough above the horizon to illuminate the surface but not overhead to cause cast shadows. If this is correct could you state it, or be more precise in what you mean. Also, in the meteorological paragraph above maybe you could say that it's common to have good weather early in the morning during this time of year, or some similar support to your saying here that the weather is almost always good.

A: We added the sentence as "Sunlight was high enough above the horizon to illuminate the surface but not overhead to cause cast shadows."

16. L88: Do you mean to say you obtained an accuracy of around 20 cm? >20cm could mean almost anything.

A: We changed sentence as "We obtained an absolute position within 30 cm accuracy at GCPs positions by post-processing with data from Kyrgyz GNSS reference station."

17. L92: Do you have any constraint on the z accuracy of your DSMs? Did you find some to be better than others in your stack, and if so, any variable that you suspect is responsible?

A: We used the standard for each lake when each lake become minimum area during survey period. In addition, we changed to elevation from DSM based on GCPs from Trimble GNSS data to make DSMs.

18. L93: This is a strange location for a reference to Fig 2. I would remove.

A: We removed and added it.

19. L94: The sentence "Workflow of SfM software was almost automatic without setting of GCPs which we measured." Is unclear and needs to be explained further.

A: We changed sentence as "Although workflow of SfM software was almost conducted automatically using the location of the images from the UAV, we set GCPs manually taken by Trimble GeoExplorer6000 in GCP setting. The DSM was used to correct relative water level fluctuations to elevation values."

20. L96: [an] ortho-image and DSM [remove: data] as the: : :

A: We changed it.

21. L96: This first step is unclear, are you shifting the DSM in only x,y space or also in some averaged z? There are competing signals of emergence, glacier melt and elevation loss as the features move down slope. With some more specific information about this glacier you may be able to make an argument to neglect these terms, e.g. your measured dz is much greater than the sum of these terms over the observation period or that the observation spatial domain is small enough that emergence at least can be assumed uniform, but they need to be addressed in some way since you compare absolute elevation data. It seems logical you would use the first image in a stack as the reference image but that isn't stated here, why not?

A: We shifted the DSM and ortho-image in only x, y space. We investigated the relative changes of water-levels based on one standard DSM for each lake. Standard DSM is different for each lake. After that, we changed to elevation change using DSM based on GCPs from Trimble GNSS data to make DSMs. In the study site (2.5 km x 2.5 km), vertical change is quite small for one month due to the glacier flows of 4m/month and average small slope gradient (<2 degree).

We improved the sentence as "Since the glacier surface flows at a speed of about 4 m/month in the study site, all other ortho-images and DSM data were fitted to the standard data by shifting them with the use of the 'georeference' function (none liner) with more than 10 tie points in ArcGIS. We just shifted the DSM in x and y space. The vertical differences for one month due to glacier motion and quite low gradient in the study site (2 x 2.5 km) is quite small."

22. L99: Was the shift to correct for glacier flow linear? Meaning did you use only one tie point or several that then caused a nonlinear translation?

A: We used more than 10 tie points to make georeferenced image for each image (none liner). We showed the sentence in question 21.

23. L101, Step 4: Are these points throughout the polygon or only along the perimeter?

A: We added words as "We converted edge line of the lake polygon to points at 15-cm intervals."

24. L103: Excluding 45% sound like a lot. How did you choose this value? You must know a lot about SfM errors over the flat surface of water, could you add some detail about the errors? An oblique angle image of the ortho-image draped over the corresponding DSM could be useful for the reader to see and understand the source data and why a 45% filter was applied.

A: We understand DSM errors on water area. We just used elevation value of edge line of lake polygon. However, edge line also includes small errors. So, we used mean average of elevation value at edge line of lake polygon. 45% is best to exclude error data, comparing to result of water level data logger. However, several hundred data remain after excluding.

25. L103-104: You mention methods here for changes in water volume and periodically report values in the text but these results are never presented on their own. I think there is a lot of potential to tell a more complete story by presenting these findings as well.

A: We already investigated the water balance between inflow and outflow (drainage) based on water volume in detail. However, water balance is not coincided each other. As the reason, volume of englacial network might be larger or the connecting network of supraglacial lakes might be wider area. Thus, water volume is needed under more consideration. We try to analyze more for next paper.

26. L105-107: Lake area variations are not shown in any table or figure, I can see this being a useful measure to bridge the unknown periods between your windows of surface elevation measurements but this is not shown in your results, nor a discussion of how lake area and dz are coupled.

A: We calculated water balance using area. We focused on water level, because water balance is not coincided with inflow and outflow. In addition, lake are variations are so small due to steep topography in study site. Water volume and area is needed under more consideration. We try to analyze more for next paper.

27. L111-114: All field glaciologist will empathize with deployed sensors not returning useful data, but I don't think it's necessary to describe here. If the sensors in lakes 4 and 10 did not record any useful data, then it would ease the reading of this section to remove them from this paragraph altogether.

A: We removed it.

28. L125: Here you say the larger lakes are in the central part of the glacier, I assume you mean orthogonal to flow, not along flow, but this is not clear.

A: We changed the sentence as "The largest of these supraglacial lakes (>5000 m2) were located in the central part along glacier flow direction in study site."

29. L127: I like the organization and discussion of drainage events, however I think there are still some bulk results that you can make here. Specifically, more comprehensive results of volume changes and maybe analysis of the spatial distribution of draining and not draining ponds as you show in the figures with colored numbers.

A: As shown in question 25, we already investigated water balance with relationship between inflow and outflow (drainage) based on water volume, because our aim is to check water balance before. However, water balance is not coincided each other. We want to consider as the subject next paper. We already showed the spatial distribution of drainage and without drainage ponds based on different timing (for example, Fig. 12b).

30. L129: "Consider the storage and drainage behavior of the nine lakes marked in Fig. 3a" Is an unnecessary sentence. Consider starting this section with a statement.

A: We changed the sentence as "The water-level elevations of the nine lakes (Nos. 1, 8, 9, 10, 11, 12, 13, 14, 16; Fig. 3a) in summer 2017 range from 3307 to 3321 m, but each one varies during 22 July to 15 August 2017 (Fig. 3b)."

31. L131 and throughout: ": : :increased gradually by 0.3 m (2723 mˆ3)" Is this a rate per day? Please specify the time here. And considering sidewall slope, the volume will be different for identical surface changes or losses. Consider reporting an integrated change rather than a rate or be clear with assumptions/averages.

    A: This is integrated value of change for three days from 22 to 25 July. We changed the sentence as "For example, the water level of the largest lake (No. 8) increases by 0.3 m (2723 m3) from 22 to 25 July 2017,,,,"

32. L136: Did you measure lake area change? It would be interesting to see a plot of lake area change vs volume change and I believe you have all the data to plot this. A statistically significant relation would make some inference on lake ice wall geometry.

    A: We have the data of area and volume. However, as written before, we could not get good water balance between inflow and outflow. We do not understand using of volume and area data without form of basin. However, we would like to try area and volume variations for next paper.

33. L136: "a conduit with exposed parts" this is a recurring concept throughout this article that seems confusing to me. In my mind a conduit in glaciology is an englacial channel, while surface streams can thermally erode channels and sometimes briefly go under ice bridges, but I wouldn't call them a conduit. It's fine that you use terms differently, and perhaps my view is incorrect, but maybe in the introduction you could write a short section or few sentences describing how you define a conduit.

    A: We added the sentence which defined englacial and supraglacial conduit as "We defined here englacial conduit as a conduit covered on all sides by ice and debris, and supraglacial conduit as a conduit with some exposed surface."

34. L137: "[That exposure parts, change to: These ice exposed parts] were caused by erosion from the drainage water: : :" I assume you mean erosion of both ice and debris cover, if so, please state that to be more clear.

    A: We changed the sentence as "New L148-149: These ice exposed parts were caused by erosion of ice and debris cover from the drainage water flowing from lake 8 (Fig. 4b)."

35. 142-143: You write that lake 1 had a gradual increase of water, but in the sentence before you said it rose by 7.5 m in one day before the next drainage. This seems like a rapid refill to me, what do you mean exactly by gradual?

A: As pointed out, lake 1 showed the raid refill after drainage. After that, the level became a gradual increase just before drainage. We changed the sentence as "Although Lake 1 showed the rapid refill after drainage, the lake level became to a gradual increase just before drainage. All three lakes (1, 8, 9) show a gradual increase of water level just before drainage."

36. L145: For the lakes that maintain a stable water level do you have evidence, e.g. from satellite data, that they are also draining just not during this period? Can you say if any lakes persist throughout the course of a year or more?

    A: Some lakes remains more than one year. We changed the sentence as "The remaining lakes (Nos. 10, 11, 12, 13, 14) maintain a stable water-level during the survey period. Two lakes (10, 11) were not drained completely more than one year."

37. L146: This section is essentially a figure caption. I think you should bring out more of the significance of why this event is one of your key results or consider merging it to a more general section listing events and citing figures. One key result that seems missing from this event is a plot of logger vs DSM derived change in water surface elevation.

    A: We merged 4.2.2 and 4.2.3, and change title. We added two sentences and supplemental file of water level data from data logger data and DSM as "Temperature data remains around 1-2 ºC from the time that water level starts rising above the data logger." And "These results show simultaneous drainage events from several lakes occurred on 12-13 May and 17-18 July in the range of at least 1 km$^2$."

38. L165: rewrite: increased by 0.6-2.0 m$^{-1}$

    A: This increase value is total, not rate per day. We added the date as "Before the drainage event, the water levels of lakes 1, 4, 7, and 8 gradually increased by 0.6–2 m from 8 to 17 July."

39. L167: For the lake(s) that gradually filled and then maintained a stable level, what can you infer about the water table of the glacier at this location and the bulk permeability?

    A: We discussed in the last chapter in DISCUSSION. It shows that the water level in each lake connected to network. Water table has reached a stable state for sharing water.

40. L169: It's great that you use satellite imagery to complement your timeseries but it leaves us guessing when you were monitoring this region. Could you please add a timeline figure or subfigure that shows the sensor, datatype and timing of each acquisition you used for this analysis.

    A: We added the time table.

41. L172: Change to "This result suggests the drainage route was: : :" and later "lakes 14 and 15 after

the 13 May.."

A: We changed the sentence as "This result suggests the drainage route was changed to the direction of lakes 14 and 15 after the 13 May drainage event."

42. L174: "For [the] drainage.."

A: We changed it.

43. L174: That the "conduit" is visible in Fig. 8 means it is a supraglacial stream or channel not englacial.

A: We changed the sentence as "the water flowed through a mixed-complicated conduit of englacial and supraglacial parts."

44. L176: The sentence starting "The drainage route with exposed ice wall: : :" Should be rewritten more clearly, I think your key point is the water took a different drainage path but the positioning relative to ice walls is unclear.

A: We showed drainage direction of July 2018 in figure 5. We changed the sentence as "The position of the exposed ice-walls is was different in May and July (Figs. 5 and 8)."

45. L177: Consider: "This difference indicates a change in drainage route between"

A: We changed the sentence as "This difference indicates a drainage route changed between May and July."

46. L182: Cite a figure and consider just describing the outcomes here that are notable, it currently reads more like a literal description of a figure.

A: We added the sentence as "Water level of these two lakes increased gradually before drainage."

47. L189: This sentence would also benefit from citing a timeline figure of data sources. Maybe above the timeline you could have the number of the lake that drained at tha time.

A: We added the number of lake on the time table.

48. L195: Why was the route unclear? There was no geomorphological change evident? Does this suggest it was a smaller event? Or are you considering the drainage went truly englacial?

A: Yes, we think surface change was not shown clearly, but we confirmed it. As the reason of unclear, this drainage event was small scale.

49. L201: "..despite being." This sentence is incomplete.

   A: We changed the sentence completely as "Seven lakes (Nos. 1, 4, 5, 7, 8, 14, 15) drained at the same time on 17 July 2018, indicating that they shared one or more englacial conduits despite being on different branch glaciers (Fig. 2c)."

50. L204: "indicating that some part of the englacial conduit system changed" I would like to have a sense for what is truly englacial and what is supraglacial.

   A: We defined here englacial conduit as a conduit covered on all sides by ice and debris, and supraglacial conduit as a conduit with some exposed surface. If there are both in the same place, we showed the water flowed through a mixed-complicated conduit of englacial and supraglacial parts.

51. L205-207: This is almost a direct quite from the introduction.

   A: These sentences are important in discussion. We changed the sentence as "The storage and drainage of supraglacial lakes is caused by closure and opening of englacial conduits (Benn et al., 2012). As factors of the closure of an englacial conduit, roof collapses, creep closure, freezing of stored water, deposition of ice and debris, and thermal erosion are reported (Sakai et al., 2000; Gully and Benn, 2009; Narama et al., 2018)."

52. L207-208: How can you be sure there wasn't always some flow through the lake? In other words, is the water truly cut of during this time, as you state here, or is there a, possibly small, but constant in and out flow while the lake level is stable?

   A: Although there is no definite proof that the tunnel was closed, we thought the amount of debris and ice that closed the tunnel was small. Because recharging and re-drainage were short period within a few days.

53. L216: give the figure number along with the letter for each reference, here and below.

   A: We changed like below.
   (a) → (Fig. 10a), (b) → (Fig. 10b)…

54. L217 Do you have any evidence suggesting why a lake would partially drain? The conduit is not at the deepest point of the lake? Or there is a blockage during the first half of draining?

   A: As the reason of partial drainage on lake 5, water level of lake 5 became stable after 12 July (Fig. 6c). The englacial conduit position was deepest point at lake 5 bottom. This evidence is shown in Fig. 7b. An englacial conduit was not closed completely. The stopping of water level decrease was caused sharing water with the others.

A: We changed the sentence as "The first drainage of lake 5 on 8 July 2018 was caused by the branch englacial conduit connected to the main englacial conduit in Fig. 10a. Although the englacial conduit is located at lake bottom (Fig. 7b), this drainage was not complete due to deposition of ice and debris. The lake's water level was maintained until the multiple drainage event on 17 July because the water of lake 5 was shared with the others (Nos. 4, 7, 8) (Fig.6c)."

55. L239: You can't really say "this observed behavior is consistent with the distribution of englacial conduits shown in Fig. 11" because you drew the conduits shown in Fig. 11. You can say "We estimated the location of conduits following the theoretical model put forth by x (citation) and guided by the behavior we observed" Or were these supraglacial channels that you could map and not infer?

A: We changed the sentence as "The location of conduits in Fig. 11 were estimated due to the observed behavior."

56. L243-245: Does conduit reuse have a limiting depth within the glacier? I assume at some depth overburden pressure will close any unpressurized channel.

A: We do not have a data of the relationship between reuse and a limiting depth.

57. L251-254: The sentence beginning " This behavior indicates that the opening" is unclear. It starts as a general statement "of an englacial conduit" but then changes to a past tense description of a particular event "maybe have been caused by." The further point relating water pressure and water volume is also not clear to me. The following sentence is also unclear, I believe the conduit would need to be pressurized to have commutative lake level fluctuations.

A: We describe here as an introductory part to consider the situation of simultaneous drainage event. This behavior shows "gradual increasing of water level and then drainage at simultaneous drainage event". In addition, the simultaneous event was caused by drainage from several lakes in the study site (at least 1km$^2$).
We exchanged the sentence as "This behavior related to gradual increasing of water level and then drainage indicates that the opening of an englacial conduit may be caused by an increase in water pressure in the conduit due to an increase in the amount of water. However, their water levels indicate different elevations before drainage (Fig. 6c). If those lakes were connected to same englacial conduit, water-levels of those lakes become the same level, like a communicating vessels."

58. L260-261: Please explain further how other studies found use for hydraulic potential and if they also found a linear relation with a similar slope. A gradient from 5 points is not very convincing, could you please find/plot this gradient for all of the multi-lake drainage events you constrain and then build on your interpretation of what this means or infers.

A: We can show another case of drainage event from 15-16 July 2018. We showed figure 13 of lake 32 drainage event. Lakes of drainage (orange point) are located on the same hydraulic gradient line, like figure 12. We also added previous study (How et al., 2017) about hydraulic potential as the same case.

We changed the sentence as "In Fig. 12c, we also added a hydraulic gradient line (orange, dashed) because recent glacier studies have found it useful to examine the hydraulic potential (Benn and Evans, 2010; How et al., 2017; Ravier and Buoncristiani, 2018). How et al. (2018) reported that subglacial channel calculated from hydraulic potential indicated connecting plume point and supraglacial lake which observed drainage."

We changed the sentence as "If so, using satellite data to monitor the variations of area and water levels of supraglacial lakes may confirm this pattern of these water storage and drainage events. In addition, understanding of these phenomena before water discharge help reduce damage caused by large-scale drainage events."

[Figure]

Fig. 12                                        Fig. 13

59. L282-283: It's not enough to say "The englacial conduit system changes over a timescale of months", you need to qualify this with a summary of what exactly changed.

A: We changed the sentence as "In addition, drainage routes of englacial conduit were changed over a timescale of months."

60. L286: Why not take this one step further and consider lake area change to infer conduit connectivity?

A: If englacial channel connection was guessed by area variations of supraglacial lakes, it would be need to observed multi periods. Although we investigated water balance, we do not have enough evidence to indicate that result. So, we want to try the issue for next paper.

61. Figure 1: I don't understand breaking the glacier into a debris covered part and the rest of the glacier along an arbitrary line, roughly mapping the debris cover for this figure would be more logical. I don't see the reason for putting Koilu Station and Tien Shan Station on this map or in this paper.

A: We need the climate condition of annual precipitation and temperature as a basic data of climate environment. Because some scientists discuss with lake area change due to increasing of precipitation.

62. Figure 2: 3 GNSS points concentrated in the lower right of panel C doesn't seem like very many ground control points. Was there a loss of DSM accuracy with increasing distance from these ground control?

A: The accuracy is not so change due to GCPs, because we also used the location of UAV. The change of water level was extracted as relative change using standard DSM for each lake. After that we converted to elevation.

63. Why did you change the lake color to blue? The red lines show which lakes you identified and any variation in turbidity captured in the color of the water itself is also interesting.

A: It is difficult to distinguish each lake, because there are many small lakes.

64. Figure 3 (and Fig 6): I like these figures a lot, however it would be slightly easier to follow if rather than random colors, the lakes and water-level elevation lines were colored in a gradient color-scheme with distance from the lowest down glacier point in the image. This would allow the reader to easily know the relative elevation ranking of each lake.

A: We changed the color.

65. Figure 3: Why was the obvious lake in the lower right excluded?

A: This lake is located 10m higher position compared to another lake. If we include this lake, we can not see small changing of lake levels. In addition, the lake did not change during the survey period.

66. Figure 4, 8 and 9: If you can see the expression of ice and rock erosion of a drainage event from a drone in the sky, it is be definition not englacial. Please correct this throughout the manuscript. There may be instances where the water is englacial, particularly at the lake itself, but I think a clear distinction needs to be made.

A: We defined here englacial conduit as a conduit covered on all sides by ice and debris, and supraglacial conduit as a conduit with some exposed surface. If these are located the same place, we showed as a mixed-complexed conduit of supraglacial and englacial parts. We showed these sentences on the text.

67. Figure 4: Since this lake does not drain completely and looking at the drainage path, it looks more like a thermally eroded ice dam that caused the partial drainage.

A: This lake formed along the englacial conduit. If the conduit developed at lake bottom, lake water was discharged completely. The englacial conduit is located middle point, because thermal erosion promotes melting of lake bottom after that. So, lake water could not discharge completely. Conduit positions are various like lake bottom or side wall.

68. Figure 5: Please add here or in a separate figure the same water level variations from both the logger data and the DSM data.

A: It is difficult to show the water level in May using DSM, because resolution of Planet and UAV DSM in July causes larger errors. Although we can add the data in July, we can show the supplemental file.

69. Figure 10: After considering the semantic, englacial vs supraglacial channel points made several times above, please re-evaluate this figure and insure it appropriately shows water below the surface only when there was in fact water flowing in channels below the surface.

A: We improved it.

70. Figure 11: Missing scale. It also might help to have a vector showing the general direction of ice flow.

A: We added the scale and a vector of glacier flow.